# Comparing Spanish-Style and Natural Fermentation Methods to Valorise Carolea, Nocellara Messinese and Leccino as Table Olives

Nicolina Timpanaro [1], Chiara A. C. Rutigliano [1], Cinzia Benincasa [2,*], Paola Foti [1,3], Solidea Mangiameli [1], Rosa Nicoletti [2], Innocenzo Muzzalupo [4] and Flora V. Romeo [1]

1    Council for Agricultural Research and Economics (CREA), Research Centre for Olive, Fruit and Citrus Crops, Corso Savoia 190, 95024 Acireale, Italy; nicolina.timpanaro@crea.gov.it (N.T.); chiara.rutigliano@outlook.it (C.A.C.R.); paola.foti@phd.unict.it (P.F.); solidea.mangiameli@gmail.com (S.M.); floravaleria.romeo@crea.gov.it (F.V.R.)
2    Council for Agricultural Research and Economics (CREA), Research Centre for Olive, Fruit and Citrus Crops, Via Settimio Severo 83, 87036 Rende, Italy; rosa.nicoletti@crea.gov.it
3    Department of Agriculture, Food and Environment, University of Catania, 95123 Catania, Italy
4    Council for Agricultural Research and Economics (CREA), Research Centre for Forestry and Wood, Via Settimio Severo 83, 87036 Rende, Italy; innocenzo.muzzalupo@crea.gov.it
*    Correspondence: cinzia.benincasa@crea.gov.it

**Abstract:** This paper presents the results of the transformation into table olives of drupes belonging to three Italian cultivars: Carolea, Leccino and Nocellara Messinese, widely used for virgin olive oil production, by using the two most common methods to produce fermented table olives: the Spanish-style method (SS) and the natural fermentation (NF). The most suitable drupes as table olives due to their flesh-to-pit ratio and high-weight fruits were Nocellara Messinese olives. From the results obtained, it was highlighted that fermentation must be improved by using a selected starter culture that can drive the fermentation by rapid acidification. In fact, the long time required by NF results in a lower pH close to the hygienic safety limit but not low enough to be considered as a stable product, while the fast fermentation obtained by treating the olives with lye solution resulted in pH values that were too high. The sugar content in all table olives was almost null, and the sensory evaluation showed that SS olives were less bitter than NF olives. Moreover, NF olive-flesh samples showed a higher amount of healthy phenolic compounds than SS olives, whose phenolic content was drastically affected by the alkaline treatment and the successive washing steps.

**Keywords:** debittering methods; dual-purpose olives; phenols; sensory analysis

## 1. Introduction

The *Olea europaea* L. plant is one of the oldest tree species whose fruit is a drupe, and its products are table olives and virgin olive oil, which have historically represented a foodstuff of high nutritional value for the inhabitants of the Mediterranean basin [1]. Table olives are widespread fermented vegetable products which possess a high nutritional value due to the content in fibres and antioxidant compounds; thus, they can be considered an important functional food [2]. Nowadays, table olives are consumed as appetisers and/or highly healthy culinary ingredients due to their low sugar content, high content of unsaturated fatty acids, vitamins and antioxidant compounds [3].

In Italy, there are more than 500 varieties which, unfortunately, can have a high number of synonyms. Each Italian region has at least one typical method of processing table olives, linked not only to local customs and traditions but also to the peculiar characteristics of the olive varieties present. The regions of southern Italy have a remarkable olive growing heritage, mainly in the existing varieties, among which only some have proven suitable as table olives [4]. The choice of appropriate variety mainly depends on the morphological and

physical–chemical properties of the fruit, on consumer preferences and on the productive cycle to be applied.

The two most common methods to produce fermented table olives are: the Spanish method for green olives and the Greek method for black olives [5]. The Spanish-style method consists of a treatment with alkaline lye (1.8–2.5%, $w/v$ NaOH) to obtain olive debittering, followed by a washing step to remove the excess alkali. Then, the olives are brined (10–13% ($w/v$) NaCl) and a spontaneous fermentation takes place [6]. In southern Italy, the industrial production of black olives, as well as that of several green olive cultivars, is carried out by spontaneous fermentation processes due to the action of the autochthonous microbiota [7–10]. The natural fermentation in brine at 8–12% ($w/v$) NaCl usually lasts 8–12 months and is driven by mixed populations of microorganisms, mainly yeasts and lactic acid bacteria [7,11].

The olive industry is facing several challenges, including plant and crop management, olive quality, production methods and health issues [12]. All table-olive producers require innovative techniques that improve performance and industrial sustainability, as well as the development of new products that specifically respond to increasingly demanding consumers. Foods with optimal nutritional characteristics, high quality and safety, improved organoleptic characteristics and reduced additives are in high demand.

The aim of this work was, therefore, the valorisation as table olives of the Italian olive cultivars widely used for virgin olive oil production. The standardisation and industrialisation of the processing best suited to each cultivar could economically improve regional production. This paper shows the results of the transformation into table olives of drupes belonging to three Italian cultivars that are classified as dual-purpose olives, i.e., suitable for both oil extraction and transformation to table olives. The olives of the cultivars Carolea (from Calabria), Leccino (from Tuscany, but now ubiquitous in Italy) and Nocellara Messinese (from Sicily) [13] are compared following two processing methods: Spanish-style and natural fermentation. In the Italian market, Carolea and Nocellara Messinese are mostly consumed as green table olives [14,15], while Leccino is usually known as a black table olive [16].

## 2. Materials and Methods

### 2.1. Olive Sampling and Processing

Olive drupes of the Leccino, Carolea and Nocellara Messinese varieties were harvested in October 2020, at a green state with less than 5% turning-colour olives, supplied by organic farms in the regions of origin, i.e., Tuscany, Calabria and Sicily, respectively. The olives of each variety were washed with tap water and processed with two methods: natural (NF) and Spanish-style (SS) fermentation.

In order to carry out the natural fermentation, olives were placed into 20-L plastic (PE) containers, then filled with 8% ($w/v$) NaCl brine previously acidified with 0.1% $v/v$ of lactic acid (85%, Sigma-Aldrich, San Louis, MO, USA) and 0.1% $w/v$ of L-ascorbic acid (Lafood, Fasano, BR, Italy). For Spanish-style processing, the olives were treated with a lye solution (2% of NaOH) and, after about 3–5 h (depending on the different varieties and fruit size), several sequential water washings were performed. At this point, olives were put into 10-L glass vessels filled with the same brine used for natural NF olives (8% of NaCl, and 0.1% of two acids). The fruit/brine ratios were 1.1.

In order to favour natural fermentation, the olives were stored at $20 \pm 1$ °C in a conditioned room equipped with a thermometer for eight months (NF) and for three months (SS). Carpological analyses were conducted on 25 fruits, randomly sampled from each olive cultivar the day after harvesting. The oil yield was determined in destoned and homogenised olive pulps by Soxhlet extraction using n-hexane for 6 h. After evaporation of the solvent, the oil content was determined gravimetrically.

## 2.2. Analyses of Brine Samples

The pH of the brines was detected at each sampling time by using a MettlerDL25 pH meter (MettlerDL25, Mettler-Toledo International Inc., Columbus, OH, USA). The olive brines were filtered through PTFE filters (0.45 μm, Millipore Merk, Darmstadt, Germany) and injected in the chromatographic system consisting of a chromatography Waters Alliance 2695 HPLC equipped with a Waters 996 photodiode-array detector (PDA) set at 280 nm and with Waters Empower software (Waters Corporation, Milford, MA, USA). The used column was a Luna C18 (250 mm × 4.6 mm i.d., 5μm, 100 Å, Phenomenex, Torrance, CA, USA) maintained in an oven at 30 °C. Separation of compounds was achieved by using an initial composition of 95% of A solution (2% acetic acid in water) and 5% of B solution (methanol) (Merk, Darmstadt, Germany). The concentration of B solution was increased to 30% in 15 min and to 70% in 25 min and then, after 2 min in isocratic conditions, the mobile phase was set at the same initial concentration in 8 min. The flow rate was 1 mL/min.

Phenolic compounds were identified by injecting the pure standards (Extrasynthése, Genay, France) of hydroxytyrosol (TyrOH), tyrosol (Tyr), oleuropein (Ole) and verbascoside (Verb), and by means of their retention time and UV–Vis spectra. All the analyses were performed in triplicate.

## 2.3. Microbiological Analyses

Decimal dilutions of each brine sample were aseptically prepared and plated on the following selective media and conditions: De Man, Rogosa and Sharpe agar (MRSA, Oxoid, Milan, Italy) with 50 mg/L Nystatin for lactic acid bacteria at 32 °C for 48 h under anaerobic conditions; plate count agar (PCA, Oxoid, Milan, Italy) for mesophilic bacteria incubated at 25 °C for 48 h; Sabouraud chloramphenicol agar (SAB, Bio-Rad, Milan, Italy) for yeasts and moulds at 25 °C for 48 h; mannitol salt agar (MSA, Oxoid, Milan, Italy) for *Staphylococcus* spp. at 32 °C for 72 h; chromogenic coliform agar base (CCA, Bibby Scharlau, Milan, Italy) for coliform bacteria at 37 °C for 24 h. The method for pathogen detection and enumeration was based on plate count according to the manufacturer's instructions. The brine analyses were conducted in triplicate and the results expressed as log CFU/mL of brine.

## 2.4. Analysis of Olive Flesh Samples

### 2.4.1. Sugar Analyses

Approximately 0.2 g of dried olive drupes was extracted with 100 mL of distilled water by sonication (15 min) and further centrifugation (10 min). The filtered supernatant solution was analysed by electrospray ionization tandem mass spectrometry (ESI-MS/MS). The extraction was conducted in triplicate.

Standard stock solutions were prepared by dissolving the sugar standards, mannose, fructose, glucose and galactose (Sigma Aldrich, St. Louis, MO, USA) in distilled water. Aliquots of these solutions were further diluted to obtain five calibration standards at concentrations ranging between 0.625 and 10 μg/mL. Calibration curves were built using a least-squares linear-regression analysis with correlation coefficients between 0.9996 and 0.9999. The accuracy and precision were evaluated at two concentrations: 3.25 and 6.50 μg/mL (recovery between 94 and 109%). Measurements were performed by using an API 4000 Q-Trap mass spectrometer (MSD Sciex Applied Biosystem, Foster City, CA, USA). The LC–MS was operated in the positive-ion mode using multiple reaction monitoring (MRM): for each analyte, the transition of the deprotonated molecular ion [Sugar-H]+ was scanned on the first quadrupole and its main fragments on the third one [Sugar-Cs]+. The experimental conditions were as follows: ion-spray voltage (IS) 5500 V; curtain gas 15 psi; temperature 100 °C; ion source gas (1) 35 psi; ion source gas (2) 45 psi; collision gas thickness (CAD) medium; the entrance potential (EP), declustering potential (DP), entrance collision energy (CE) and exit collision energy (CXP) were optimized for each transition monitored.

The analytes were separated on a Chromegabond carbohydrate column (5 μm particle size, 15 cm length and 2.1 mm i.d., PerkinElmer, West Berlin, NJ, USA) at a flow rate of 300 mL/min with an injection volume of 10 μL. The binary mobile phase consisted of acetonitrile (A) and CsCl $H_2O$ 54 μM (B). The solvents of LC/MS grade were supplied by Sigma–Aldrich (Riedel-de Haën, Seelze, Germany). The total elution time was 10 min per injection.

### 2.4.2. Phenolic Analyses

In order to perform the extraction of the phenolic compounds, 1 g of homogenized olive drupes was weighed in a 50 mL volume test tube and 20 mL of methanol was added. The mixture was homogenized by means of an ultra-turrax system (Ika, Staufen, Germany) at 8000 rpm for 1 min. To maximize the extraction process, the solution was kept under shaking in an ultrasonic bath (Fisher Scientific, Milan, Italy) in darkness for 20 min. After this period, centrifugation at 5000 rpm for 25 min at 8 °C allowed the recovery of the supernatant. Subsequently, the solvent was removed under vacuum by means of a rotary evaporator (Büchi, Cornaredo, Italy) set at 40 °C and 60 rpm. Solvent-free extracts were recovered with 2 mL of a solution of water/methanol (*v/v* 80:20), filtered through a 0.45-μm PVDF filter (Merk, Darmstadt, Germany) and analysed by HPLC-MS/MS. The extraction was conducted in triplicate.

Quantitative analyses were performed by external calibration curves built using a least-squares linear-regression analysis. For this purpose, standard stock solutions were prepared by dissolving hydroxytyrosol (TyrOH), tyrosol (Tyr), oleuropein (Ole), verbascoside (Verb), rutin (Rut), lutein (Lut), lutein-4-O-glucosides (Lut4), lutein-7-O-glucoside (Lut7), apigenin (Ap), coumaric acid (Cum), vanillic acid (Van) and diosmetin (Dios) in methanol, before further dilution with water/0.1% formic acid to obtain six calibration standards at concentrations ranging between 100 and 2000 μg/mL. The correlation coefficients of the calibration curve ranged between 0.9994 and 0.9997. The standards were purchased from Sigma–Aldrich (Riedel-de Haën, Laborchemikalien, Seelze, Germany) and Extrasynthése (Genay, France).

The crude methanol extracts containing the phenolic fraction were analysed by electrospray ionization tandem mass spectrometry (ESI-MS/MS) using an MSD Sciex Applied Biosystem API 4000 Q-Trap mass spectrometer in negative-ion mode using multiple reaction monitoring (MRM). The analytes were separated on an Eclipse XDB-C8-A HPLC column (5 μm particle size, 150 mm length and 4.6 mm i.d.) (Agilent Technologies, Santa Clara, CA, USA) at a low rate of 300 μL/min with an injection volume of 10 μL. The binary mobile phase made up of 0.1% aqueous formic acid (A) and methanol (B) (Sigma–Aldrich, Riedel-de Haën, Seelze, Germany) was gradient-programmed to increase B from 10 to 100% in 10 min, held for 2 min and ramped down to the original composition (90% A and 10% B) in 8 min. The total elution time was 20 min per injection.

### 2.4.3. Total Phenolic Content Analyses

The amount of total phenols content (TPC) was detected by spectrophotometry at 756 nm according to the Folin –Ciocalteu method [17]. For this purpose, to 0.5 mL of phenolic extract (see Section 2.4.2.), 2.5 mL ten-times-diluted Folin–Ciocalteu reagent (Labochimica, Padova, Italy) was added and vigorously vortexed for 3 min. Afterwards, 2 mL of 7.5% sodium carbonate was added, subsequently mixed for 10 s and directly incubated for 2 h at room temperature. TPC was calculated using caffeic acid as a reference compound.

### 2.5. Sensory Analysis

The sensory evaluation of the tested olives was carried out following the guidelines of the International Olive Council (COI/OT/MO No 1/Rev.3 June 2021) [18]. The IOC method establishes the essential requisite and procedures for the sensory evaluation of the taste, odour, and texture of table olives and their commercial classification.

The sensory analysis of table olives was carried out by a group of 9 experienced tasters, chosen based on their aptitude. The training period included theoretical and practical sessions held twice a week applied according to the IOC [19]. The sensory analyses were carried out in the sensory laboratory of CREA (Acireale, Italy), equipped with a specific software for the acquisition and processing of sensory data (Smart Sensory box, Smart Sensory Solutions S.r.l., Sassari, Italy), in compliance with the UNI EN ISO 8589:2014 standard [20].

The judges were trained for identification of abnormal fermentation (putrid, butyric, zapateria), for gustatory attributes (saltiness, bitterness, acidity) and for kinaesthetic or texture attributes (hardness, fibrousness, crunchiness). The judges were also trained to identify other defects of table olives such as mustiness, rancidness, cooking effect, metallic taste, etc. The judges used a scale of 10-cm length.

Standard tasting glasses, containing 3 olives with the brining liquid, were evaluated by each judge. Two samples at each session were tasted. All olive samples were served at room temperature (about 20 °C), coded with a 3-digit random number. The panellists cleansed their mouths with salt-free bread and sparkling water between each sample.

*2.6. Statistical Analysis*

The statistical software IBM SPSS Statistics for Windows, version 20 (IBM Corp., Armonk, NY, USA) was used. The statistical analysis of the obtained results was performed using a one-way analysis of variance (ANOVA) and Tukey's HSD post hoc test for means separation (significance level at $p \leq 0.05$).

## 3. Results

*3.1. Carpological Analyses and Oil Content of Olives before Treatments*

The olives of the three tested cultivars are shown in Figure 1. The picture highlights the morphological differences of the dual-purpose olive drupes.

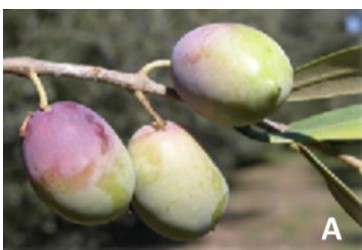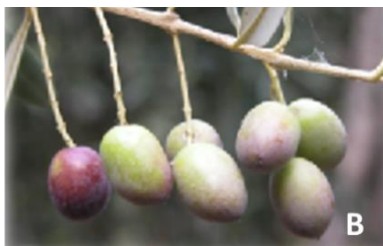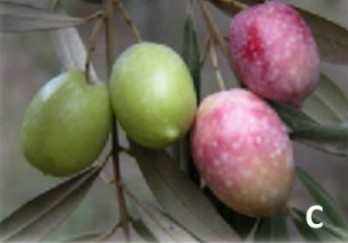

**Figure 1.** Olive branch with drupes of the three olive cultivars ((**A**) = Carolea; (**B**) = Leccino; (**C**) = Nocellara Messinese).

The carpological characteristics of the three harvested olive cultivars are shown in Table 1. Nocellara Messinese olives showed the highest carpological parameters, except for olive length obtained by Carolea drupes with 26.89 ± 2.08 mm. Leccino olives were very small fruits with a weight of less than 2 g (1.81 ± 0.36 on average). The oil yield of olive samples confirmed their green state because they did not reach the oil content of a fully ripe fruit, according to data reported by Muzzalupo [13].

*3.2. Chemical Analyses of Brines after Treatments*

The trend of the pH values and of the single phenols detected by HPLC in relation to NF treatment, are shown in Table 2. The pH of the three samples showed no statistically significant differences up to 90 days of fermentation. After 120 days of fermentation, only the Carolea samples showed pH values below the safety limit [21,22], while Leccino only slowly reached this value at the end of fermentation.

**Table 1.** Carpological characteristics of olive drupes.

|  | Carolea | Leccino | Nocellara Messinese |
|---|---|---|---|
| Olive length (mm) | 26.89 ± 2.08 | 17.56 ± 1.47 | 23.50 ± 1.58 |
| Olive diameter (mm) | 19.48 ± 1.50 | 12.74 ± 0.99 | 31.45 ± 2.33 |
| Fruit weight (g) | 5.72 ± 1.17 | 1.81 ± 0.36 | 9.14 ± 1.22 |
| Flesh weight (g) | 4.72 ± 1.06 | 1.22 ± 0.28 | 7.86 ± 1.16 |
| Stone weight (g) | 1.00 ± 0.23 | 0.59 ± 0.11 | 1.28 ± 0.24 |
| Flesh/fruit (%) | 82.5 | 67.4 | 86.0 |
| Flesh/pit | 4.8 | 2.1 | 6.4 |
| Oil yield (%) | 38.44 ± 1.32 | 34.20 ± 0.72 | 33.16 ± 0.84 |

**Table 2.** Results of pH and single phenolic compounds (mg/L) in the natural fermented (NF) olive brines.

| Cultivar | Time | pH | TyrOH | Tyr | Ole | Verb |
|---|---|---|---|---|---|---|
| Carolea |  | 4.17 ± 0.16 | 677.9 ± 171.1 a | 47.2 ± 9.3 a | 808.2 ± 115.3 a | 743.0 ± 176.4 a |
| Leccino | 15 | 4.52 ± 0.03 | 274.2 ± 21.7 b | 36.4 ± 5.0 ab | 0.0 ± 0.0 b | 23.7 ± 10.4 b |
| N. Messinese |  | 4.50 ± 0.01 | 301.8 ± 2.3 b | 25.9 ± 2.9 b | 86.2 ± 6.8 b | 37.2 ± 1.9 b |
| Sig. |  | n.s. | ** | ** | ** | ** |
| Carolea |  | 4.43 ± 0.06 | 1490.4 ± 80.3 a | 83.6 ± 6.2 a | 928.7 ± 195.6 a | 1278.1 ± 132.4 a |
| Leccino | 30 | 4.68 ± 0.17 | 1021.7 ± 226.2 b | 59.1 ± 3.5 b | 151.9 ± 105.5 b | 49.8 ± 3.6 b |
| N. Messinese |  | 4.57 ± 0.01 | 555.2 ± 17.5 c | 36.9 ± 3.9 c | 0.0 ± 0.0 b | 53.0 ± 2.3 b |
| Sig. |  | n.s. | ** | ** | ** | ** |
| Carolea |  | 4.64 ± 0.02 | 2179.5 ± 175.7 a | 84.6 ± 6.2 | 692.9 ± 12.5 a | 1750.9 ± 151.3 a |
| Leccino | 60 | 4.68 ± 0.02 | 1331.7 ± 84.5 b | 84.1 ± 7.4 | 0.0 ± 0.0 b | 15.7 ± 10.3 c |
| N. Messinese |  | 4.67 ± 0.02 | 1214.0 ± 16.5 b | 86.3 ± 6.0 | 0.0 ± 0.0 b | 118.0 ± 6.6 b |
| Sig. |  | n.s. | ** | n.s. | ** | ** |
| Carolea |  | 4.94 ± 0.09 | 2616.8 ± 250.3 a | 121.7 ± 3.2 a | 432.3 ± 47.0 a | 2078.5 ± 266.4 a |
| Leccino | 90 | 4.55 ± 0.15 | 1420.0 ± 188.5 c | 97.5 ± 8.8 b | 0.0 ± 0.0 b | 56.7 ± 3.4 c |
| N. Messinese |  | 4.70 ± 0.08 | 2003.1 ± 27.5 b | 0.0 ± 0.0 c | 0.0 ± 0.0 b | 177.6 ± 14.4 b |
| Sig. |  | n.s. | ** | ** | ** | ** |
| Carolea |  | 4.43 ± 0.07 b | 3076.8 ± 279.0 a | 0.0 ± 0.0 | 0.0 ± 0.0 | 2381.2 ± 161.2 a |
| Leccino | 120 | 4.71 ± 0.04 a | 1854.6 ± 122.6 b | 0.0 ± 0.0 | 0.0 ± 0.0 | 98.0 ± 19.2 b |
| N. Messinese |  | 4.60 ± 0.01 ab | 1900.8 ± 118.0 b | 0.0 ± 0.0 | 0.0 ± 0.0 | 183.9 ± 11.0 b |
| Sig. |  | * | ** | n.s. | n.s. | ** |
| Carolea |  | 4.41 ± 0.04 b | 3370.6 ± 178.2 a | 0.0 ± 0.0 | 0.0 ± 0.0 | 2566.6 ± 70.8 a |
| Leccino | 150 | 4.77 ± 0.02 a | 1715.7 ± 61.7 c | 0.0 ± 0.0 | 0.0 ± 0.0 | 84.2 ± 13.1 c |
| N. Messinese |  | 4.65 ± 0.04 a | 2132.8 ± 53.8 b | 0.0 ± 0.0 | 0.0 ± 0.0 | 216.6 ± 3.9 b |
| Sig. |  | ** | ** | n.s. | n.s. | ** |
| Carolea |  | 4.45 ± 0.01 b | 3547.8 ± 214.7 a | 0.0 ± 0.0 | 0.0 ± 0.0 | 2716.7 ± 238.6 a |
| Leccino | 180 | 4.66 ± 0.00 a | 1989.1 ± 87.1 c | 0.0 ± 0.0 | 0.0 ± 0.0 | 98.8 ± 9.2 b |
| N. Messinese |  | 4.69 ± 0.06 a | 2363.3 ± 112.4 b | 0.0 ± 0.0 | 0.0 ± 0.0 | 244.1 ± 17.4 b |
| Sig. |  | ** | ** | n.s. | n.s. | ** |
| Carolea |  | 4.36 ± 0.05 b | 3944.1 ± 189.4 a | 0.0 ± 0.0 | 0.0 ± 0.0 | 2924.0 ± 204.5 a |
| Leccino | 210 | 4.60 ± 0.02 ab | 1868.5 ± 29.7 c | 0.0 ± 0.0 | 0.0 ± 0.0 | 97.3 ± 13.2 b |
| N. Messinese |  | 5.02 ± 0.27 a | 2498.9 ± 168.8 b | 0.0 ± 0.0 | 0.0 ± 0.0 | 224.8 ± 5.5 b |
| Sig. |  | * | ** | n.s. | n.s. | ** |
| Carolea |  | 4.35 ± 0.07 b | 4378.6 ± 99.6 a | 0.0 ± 0.0 | 0.0 ± 0.0 | 3171.2 ± 94.7 a |
| Leccino | 240 | 4.55 ± 0.06 ab | 1998.7 ± 73.4 c | 0.0 ± 0.0 | 0.0 ± 0.0 | 92.0 ± 7.3 c |
| N. Messinese |  | 4.96 ± 0.17 a | 2741.8 ± 82.9 b | 0.0 ± 0.0 | 0.0 ± 0.0 | 258.6 ± 3.6 b |
| Sig. |  | * | ** | n.s. | n.s. | ** |

Data are expressed as means ± standard deviations. Different letters indicate statistical differences within the same column and for each sampling time. ** Significance at $p \leq 0.01$; * Significance at $p \leq 0.05$; n.s. not significant.

Regarding phenols, TyrOH values always increased and were statistically different, with the highest values reached for Carolea samples at each sampling time, and the lowest rate of increase for Leccino samples. Tyr also increased in all samples up to 90 days, after which it was no longer found in the brines. Ole was found regularly only in Carolea samples with a decreasing trend, while for the other cultivars, it was found only in the

first two samplings. Verb highlighted a trend similar to that of TyrOH, but in this case, it reached much higher values in the Carolea samples, more than 30-fold compared to Leccino and more than 12-fold compared to N. Messinese at the end of sampling.

Regarding SS olives, the pH value was always above the hygienic safety value (>4.5) with the highest average in N. Messinese brines (Table 3). The olives treated with this method were already debittered after two months due to the applied lye solution. The alkali treatment also affected the phenolic content; accordingly, the single phenols showed much lower concentrations than the same samples treated with the NF method. With SS treatment, TyrOH and Tyr contents were similar for Carolea and N. Messinese at certain sampling times. The phenolic content of Leccino brines was the lowest in this case also. Ole was never detected, while Verb was always higher in Carolea samples than in the other two cultivars, although at lower values than in the same samples treated with the NF method.

**Table 3.** Results of pH and single phenolic compounds (mg/L) in the Spanish-style (SS) olive brines.

| Cultivar | Time | pH | TyrOH | Tyr | Ole | Verb |
|---|---|---|---|---|---|---|
| Carolea | | 5.66 ± 0.14 | 1322.9 ± 226.0 b | 69.4 ± 6.8 b | 0.0 ± 0.0 | 217.9 ± 45.8 a |
| Leccino | 7 | 5.61 ± 0.01 | 973.8 ± 124.8 b | 56.6 ± 6.2 b | 0.0 ± 0.0 | 0.0 ± 0.0b |
| N. Messinese | | 5.87 ± 0.40 | 1990.1 ± 396.6 a | 126.2 ± 17.7 a | 0.0 ± 0.0 | 0.0 ± 0.0 b |
| Sig. | | n.s. | ** | ** | n.s. | ** |
| Carolea | | 5.29 ± 0.03 b | 2399.1 ± 76.1 b | 136.8 ± 5.3 a | 0.0 ± 0.0 | 776.4 ± 34.0 a |
| Leccino | 15 | 5.19 ± 0.02 b | 1534.5 ± 44.9 c | 82.3 ± 2.9 b | 0.0 ± 0.0 | 63.5 ± 2.6 b |
| N. Messinese | | 6.17 ± 0.03 a | 2668.2 ± 100.6 a | 144.5 ± 18.7 a | 0.0 ± 0.0 | 29.4 ± 12.9 b |
| Sig. | | ** | ** | ** | n.s. | ** |
| Carolea | | 5.02 ± 0.02 b | 2768.3 ± 188.6 a | 148.8 ± 14.7 a | 0.0 ± 0.0 | 1443.3 ± 140.5 a |
| Leccino | 30 | 4.86 ± 0.03 b | 1583.6 ± 99.0 b | 75.4 ± 6.0 b | 0.0 ± 0.0 | 103.1 ± 5.7 b |
| N. Messinese | | 5.71 ± 0.15 a | 2826.9 ± 340.1 a | 150.7 ± 15.8 a | 0.0 ± 0.0 | 111.4 ± 9.7 b |
| Sig. | | ** | ** | ** | n.s. | ** |
| Carolea | | 5.10 ± 0.01 | 2580.2 ± 261.1 a | 155.1 ± 4.9 a | 0.0 ± 0.0 | 1793.4 ± 118.0 a |
| Leccino | 60 | 5.58 ± 0.20 | 1368.7 ± 166.7 b | 95.6 ± 4.6 c | 0.0 ± 0.0 | 132.4 ± 7.8 b |
| N. Messinese | | 5.58 ± 0.21 | 2848.5 ± 331.3 a | 134.4 ± 8.7 b | 0.0 ± 0.0 | 165.2 ± 7.3 b |
| Sig. | | n.s. | ** | ** | n.s. | ** |

Data are expressed as means ± standard deviations. Different letters indicate statistical differences within the same column and for each sampling time. ** Significance at $p \leq 0.01$; n.s. not significant.

### 3.3. Results of Olive Flesh Samples

3.3.1. Results of Sugar Analysis

The highest content of total sugars, summing mannose, fructose, glucose and galactose, was recorded in N. Messinese fresh olives (18.67 g/100 g dry weight (DW)) followed by the Leccino (12.51 g/100g DW) and Carolea (10.59 g/100 g DW) varieties. At the end of the transformation into table olives, these values were greatly reduced to 0.09 g/100 g DW for Leccino, 0.81 g/100 g DW for N. Messinese and 0.45 g/100 g DW for Carolea (Spanish-style method). Only N. Messinese natural fermented olives gave a value of 0.065 g/100 g DW of sugars. In fact, the sugar content in the Leccino and Carolea samples was practically null (Table 4).

3.3.2. Results of Phenolic Analysis

All the results are summarized in Table 4. The fresh olive pulps were very rich in phenolic compounds: the highest content of total phenols was registered in Carolea olives (10,734 mg/kg), followed by N. Messinese and Leccino olives (8186 mg/kg and 7824 mg/kg, respectively). These values, at the end of the fermentation processes, had undergone a significant decrease. In particular, the olives processed with the Spanish method had suffered a loss of phenols of more than 90% and those subjected to natural fermentation had lost 70%. The main phenol compounds present in fresh olive pulps were Ole, Verb, TyrOH and Tyr. The richest cultivar in Ole was Carolea (3189 mg/kg), followed by N. Messinese (2061 mg/kg) and Leccino (2015 mg/kg). At the end of the

transformation with the Spanish method, Ole values were 31, 37 and 17 mg/kg, for Carolea, N. Messinese and Leccino, respectively. These values correspond to less than 1% of their initial values. Sodium hydroxide, in fact, cleaves this compound, resulting in an increase in TyrOH concentrations. Moreover, the table olives registered a final value of TyrOH of 502 mg/kg for Carolea, 464 mg/kg for Leccino and 429 mg/kg for N. Messinese. An increase was also recorded for Tyr: from an initial value of 113 mg/kg, it reaches 274 mg/kg for Carolea; from an initial value of 104 mg/kg, it reaches 294 mg/kg for Leccino; and from an initial value of 52 mg/kg, it reaches 126 mg/kg for N. Messinese. This contrasts with the trend of table olives naturally fermented, where Ole was, in any case, recovered at about 25% of its initial value. TyrOH and Tyr concentrations were also found to be higher and greater than in table olives fermented with the Spanish method.

**Table 4.** Results of single and total phenols (mg/kg) and soluble sugars (g/100 g DW) in olive pulps.

| Cultivar | Time | Total Phenols | Total Sugars | Ole | Tyr | TyrOH | Verb | Van |
|---|---|---|---|---|---|---|---|---|
| Carolea | | 10,734.2 ± 309.6 a | 10.59 ± 2.12 b | 3189.2 ± 87.9 a | 113.0 ± 3.8 a | 506.5 ± 25.4 a | 2318.3 ± 46.5 a | 139.4 ± 8.2 a |
| Leccino | 0 | 7824.2 ± 222.5 b | 12.51 ± 2.02 b | 2015.2 ± 39.3 b | 104.1 ± 5.5 a | 321.3 ± 7.2 b | 1221.2 ± 47.7 b | 26.74 ± 1.52 b |
| N. Messinese | | 8185.9 ± 290.1 b | 18.67 ± 2.55 a | 2060.6 ± 79.6 b | 51.54 ± 1.38 b | 258.6 ± 9.6 b | 2112.6 ± 94.3 a | 32.97 ± 8.84 b |
| SS | Sig. | ** | * | ** | ** | ** | ** | ** |
| Carolea | 90 | 1109.9 ± 25.7 a | 0.45 ± 0.15 ab | 31.08 ± 0.22 b | 273.6 ± 6.8 a | 502.1 ± 1.8 a | 63.46 ± 3.98 a | 71.15 ± 0.39 a |
| Leccino | 60 | 967.2 ± 71.7 ab | 0.09 ± 0.01 b | 16.94 ± 1.32 c | 293.8 ± 6.4 a | 464.4 ± 16.9 ab | 17.76 ± 0.36 b | 7.65 ± 0.31 b |
| N. Messinese | | 839.6 ± 71.6 b | 0.80 ± 0.12 a | 36.91 ± 1.29 a | 125.7 ± 9.6 b | 429.2 ± 20.3 b | 25.00 ± 4.85 b | 20.54 ± 8.83 b |
| NF | Sig. | * | * | ** | ** | * | ** | ** |
| Carolea | 370 | 3023.3 ± 13.5 | 0.0 ± 0.0 b | 656.3 ± 9.9 a | 443.8 ± 8.8 b | 561.0 ± 9.5 c | 337.65 ± 7.32 a | 115.12 ± 11.24 a |
| Leccino | 240 | 2090.4 ± 12.8 | 0.0 ± 0.0b | 517.0 ± 9.9 b | 613.6 ± 9.1 a | 684.8 ± 10.1 a | 93.77 ± 3.56 b | 37.04 ± 26.81 b |
| N. Messinese | | 2661.5 ± 13.1 | 0.065 ± 0.011 a | 535.5 ± 9.5 b | 303.5 ± 7.2 c | 628.1 ± 9.2 b | 114.1 ± 4.3 b | 27.68 ± 8.84 b |
| | Sig. | n.s. | ** | ** | ** | ** | ** | * |
| Cultivar | Time | Lut | Lut7 | Lut4 | Rut | Ap | Dios | Cum |
| Carolea | | 64.43 ± 2.05 a | 38.32 ± 3.31 b | 46.23 ± 1.26 a | 250.6 ± 70.2 | 7.21 ± 0.47 | 2.28 ± 0.42 | 8.11 ± 0.37 b |
| Leccino | 0 | 51.83 ± 1.98 b | 58.92 ± 6.97 b | 9.94 ± 0.41 b | 126.8 ± 3.3 | 7.12 ± 0.15 | 2.12 ± 0.15 | 1.40 ± 0.09 c |
| N. Messinese | | 56.02 ± 1.99 b | 98.92 ± 10.47 a | 8.31 ± 1.43 b | 103.1 ± 9.2 | 6.98 ± 0.02 | 2.16 ± 0.25 | 10.71 ± 0.41 a |
| SS | Sig. | * | ** | ** | n.s. | n.s. | n.s. | ** |
| Carolea | 90 | 43.15 ± 3.66 | 17.08 ± 1.04 | 11.36 ± 1.27 a | 6.07 ± 0.25 b | 6.66 ± 0.64 | 2.01 ± 0.02 | 9.67 ± 0.49 a |
| Leccino | 60 | 42.25 ± 2.39 | 42.93 ± 3.12 | 0.0 ± 0.0 b | 18.44 ± 1.78 a | 6.01 ± 0.21 | 1.44 ± 0.62 | 4.84 ± 0.52 b |
| N. Messinese | | 50.86 ± 6.62 | 62.49 ± 4.00 | 11.07 ± 0.57 a | 6.62 ± 0.71 b | 6.19 ± 0.27 | 2.02 ± 0.18 | 1.42 ± 0.60 c |
| NF | Sig. | n.s. | n.s. | ** | * | n.s. | n.s. | ** |
| Carolea | 370 | 39.31 ± 2.94 | 21.54 ± 1.43 ab | 17.14 ± 1.03 | 23.75 ± 8.39 | 13.13 ± 0.88 | 8.27 ± 0.42 | 20.24 ± 8.23 |
| Leccino | 240 | 37.51 ± 2.91 | 16.51 ± 1.11 b | 12.84 ± 0.87 | 31.31 ± 8.43 | 13.25 ± 0.99 | 8.25 ± 0.46 | 12.96 ± 7.96 |
| N. Messinese | | 39.75 ± 2.97 | 57.27 ± 3.32 a | 16.84 ± 1.24 | 32.58 ± 7.41 | 12.64 ± 2.83 | 8.25 ± 0.46 | 9.03 ± 6.53 |
| | Sig. | n.s. | * | n.s. | n.s. | n.s. | n.s. | n.s. |

Data are expressed as means ± standard deviations. Different letters indicate statistical differences within the same column and for each sampling time. ** Significance at $p \leq 0.01$; * Significance at $p \leq 0.05$; n.s. not significant.

The richest cultivar in Verb was Carolea (2318 mg/kg), followed by N. Messinese (2113 mg/kg) and Leccino (1221 mg/kg). At the end of the transformation with the SS, only about 3% of Verb remains in Carolea, and 1.5% in Leccino and N. Messinese table olives. At the end of the natural fermentation, in contrast, about 15% of Verb remains in Carolea table olives, 8% in Leccino and 5% in N. Messinese.

SS lowers the initial value of luteolin recorded in fresh olives by about 30% for Carolea, 20% for Leccino, and 10% for N. Messinese. Natural fermentation lowers the initial values by 10% more for each variety. Similar to all the phenols discussed above, the concentrations of Lut7 and Lut4 also underwent changes after the fermentation processes, but their trends were not entirely regular. In particular, the values of Lut7, both in SS and NF table olives, were found to be lower than the values in fresh olives. After SS treatment, in fact, about 44% of Lut7 was recovered in Carolea, 73% in Leccino and 63% in N. Messinese olives. In NF, the percentages recovered were 55, 27 and 57%, respectively. Moreover, it was interesting to note the increase in Lut4 in olives which were naturally fermented.

The richest cultivar in Rut was Carolea (251 mg/kg), followed by Leccino (127 mg/kg) and N. Messinese (103 mg/kg). At the end of SS transformation, about 2% of the initial Rut value was recorded in Carolea table olives and about 14 and 6% in Leccino and N. Messinese table olives, respectively. Rut in naturally fermented olives was about 9% of

its initial value in Carolea table olives and about 25 and 32% in Leccino and N. Messinese cultivars, respectively.

Van concentration at the end of SS transformation was about 50% of its initial value in Carolea table olives and about 28 and 60% in Leccino and N. Messinese table olives, respectively. Van in naturally fermented olives was about 80% of its initial value for the Carolea and N. Messinese cultivars; the Leccino cultivar, instead, showed an increase of 137%.

Furthermore, the value of Cum in both SS and NF table olives was found to be greatly increased.

Moreover, the concentrations of Ap and Dios were found not to be statistically significant in differentiating the cultivars, according to the two different fermentation methods. However, the trends of these phenols remain constant in both fresh and SS olives, while they are doubled in NF table olives.

### 3.4. Results of Microbiological Analysis

The microbiological analyses (Table 5) highlighted that yeasts and moulds were statistically different only at 15 and 150 days of fermentation, reaching the highest value in Carolea samples at the 90-day sampling time (6.80 Log CFU/mL). Yeasts and moulds showed a relatively constant population during NF. The lactic acid bacteria (LAB) reached the highest values in Leccino samples starting from 60 days. Additionally, in this case, LAB exhibited a constant trend during fermentation, without major increases. The total viable mesophilic count showed the maximum peak at 90 days for Carolea and Leccino and then remained stable, while in N. Messinese, this population was constant and decreased after 180 days. Coliform bacteria and staphylococci were found sporadically in a few samples at a low level, ranging from 0.90 to 2.58 log CFU/mL and from 1.00 to 3.98 log CFU/mL, respectively.

**Table 5.** Microbial log counts of yeasts and moulds (SAB), lactic acid bacteria (MRS), total viable count (PCA), staphylococci (MSA) and coliform bacteria (CCA) growth in naturally fermented olive brines.

| Cultivar | Time | SAB | MRS | PCA | MSA | CCA |
|---|---|---|---|---|---|---|
| Carolea | | 4.95 ± 0.84 ab | 5.23 ± 0.60 a | 4.07 ± 1.67 | <DL | <DL b |
| Leccino | 15 | 5.53 ± 0.83 a | 4.82 ± 0.34 a | 5.55 ± 0.14 | <DL | <DL b |
| N. Messinese | | 3.57 ± 1.17 b | 3.07 ± 0.58 b | 3.80 ± 1.32 | <DL | 1.55 ± 0.64 a |
| Sig. | | * | ** | n.s. | n.s. | ** |
| Carolea | | 5.83 ± 0.41 | 3.57 ± 0.55 | 6.20 ± 0.14 a | <DL | 2.45 ± 0.56 a |
| Leccino | 30 | 5.85 ± 0.50 | 4.09 ± 1.69 | 5.70 ± 0.42 ab | <DL | <DL b |
| N. Messinese | | 5.37 ± 0.41 | 4.83 ± 0.44 | 5.16 ± 0.49 b | <DL | <DL b |
| Sig. | | n.s. | n.s. | * | n.s. | ** |
| Carolea | | 5.74 ± 0.14 | 4.50 ± 0.70 | 5.70 ± 0.16 | <DL b | 2.60 ± 1.50 a |
| Leccino | 60 | 5.22 ± 0.51 | 5.11 ± 0.35 | 5.48 ± 0.89 | <DL b | <DL b |
| N. Messinese | | 5.00 ± 0.77 | 5.02 ± 0.72 | 5.38 ± 0.54 | 1.00 ± 0.58 a | <DL b |
| Sig. | | n.s. | n.s. | n.s. | ** | ** |
| Carolea | | 6.80 ± 0.83 | 5.84 ± 0.05 ab | 6.48 ± 1.47 ab | <DL b | <DL b |
| Leccino | 90 | 5.85 ± 1.63 | 6.28 ± 0.75 a | 7.10 ± 0.25 a | 2.25 ± 0.24 a | 2.58 ± 0.56 a |
| N. Messinese | | 5.11 ± 0.33 | 5.02 ± 0.30 b | 5.26 ± 0.33 b | <DL b | <DL b |
| Sig. | | n.s. | ** | * | ** | ** |
| Carolea | | 5.14 ± 0.29 | 4.95 ± 0.25 b | 5.16 ± 0.38 | <DL | <DL |
| Leccino | 120 | 5.42 ± 0.85 | 6.08 ± 0.34 a | 5.81 ± 0.75 | <DL | <DL |
| N. Messinese | | 5.05 ± 0.28 | 4.80 ± 0.23 b | 5.18 ± 0.98 | <DL | <DL |
| Sig. | | n.s. | ** | n.s. | n.s. | n.s. |
| Carolea | | 5.05 ± 0.36 b | 4.43 ± 0.60 b | 4.65 ± 0.54 b | <DL | <DL |
| Leccino | 150 | 6.22 ± 0.08 a | 6.32 ± 0.38 a | 6.65 ± 0.44 a | <DL | <DL |
| N. Messinese | | 5.40 ± 0.83 ab | 4.96 ± 0.63 b | 5.20 ± 0.76 b | <DL | <DL |
| Sig. | | * | ** | ** | n.s. | n.s. |
| Carolea | | 5.64 ± 0.96 | 4.80 ± 0.15 | 5.00 ± 0.00 | <DL | <DL |
| Leccino | 180 | 6.16 ± 0.48 | 5.54 ± 0.69 | 5.76 ± 0.31 | <DL | <DL |
| N. Messinese | | 4.96 ± 0.52 | 4.64 ± 0.51 | 5.00 ± 0.67 | <DL | <DL |

**Table 5.** *Cont.*

| Cultivar | Time | SAB | MRS | PCA | MSA | CCA |
|---|---|---|---|---|---|---|
| Sig. | | n.s. | n.s. | n.s. | n.s. | n.s. |
| Carolea | | 4.56 ± 0.84 | 4.72 ± 0.25 ab | 4.54 ± 0.42 | <DL | 1.65 ± 0.52 b |
| Leccino | 210 | 5.70 ± 0.16 | 5.34 ± 0.53 a | 5.11 ± 0.69 | <DL | <DL c |
| N. Messinese | | 5.26 ± 0.60 | 4.52 ± 0.26 b | 4.87 ± 0.43 | <DL | 3.38 ± 0.53a |
| Sig. | | n.s. | * | n.s. | n.s. | ** |
| Carolea | | 4.68 ± 0.22 b | 4.35 ± 0.35 | 4.67 ± 0.35 b | 3.37 ± 0.48 a | <DL b |
| Leccino | 240 | 5.29 ± 0.14 a | 4.42 ± 0.34 | 5.29 ± 0.33 a | 3.98 ± 1.49 a | <DL b |
| N. Messinese | | 4.50 ± 0.17 b | 4.11 ± 0.34 | 3.34 ± 0.43 c | <DL b | 0.90 ± 0.52 a |
| Sig. | | n.s. | n.s. | ** | ** | ** |

Data are expressed as means ± standard deviations of Log CFU/mL. Different letters indicate statistical differences within the same column and for each sampling time. ** Significance at $p \leq 0.01$; * Significance at $p \leq 0.05$; n.s. not significant. DL = detection limit.

The results of microbiological analysis obtained for SS samples (Table 6) highlighted a higher presence of yeasts and moulds in the SS samples than in the NF samples, with the highest values in Carolea and Leccino brines. Lactic acid bacteria showed a different behaviour in the three samples: it reached the highest values in Carolea samples at the end of fermentation, while Leccino showed the lowest concentration, particularly at 30 days. N. Messinese samples, instead, showed no LAB at the first sampling and then they grew and were stable up to the end of brining. The total viable mesophilic count showed stable and similar growth values among the samples. Staphylococci were found only in Leccino at the last two sampling times, while coliform bacteria were present in all the tested samples in a range between 0.33 and 5.03 log CFU/mL.

**Table 6.** Microbial log counts of yeasts and moulds (SAB), lactic acid bacteria (MRS), total viable count (PCA), staphylococci (MSA) and coliform bacteria (CCA) growth in Spanish-style olive brines.

| Cultivar | Time | SAB | MRS | PCA | MSA | CCA |
|---|---|---|---|---|---|---|
| Carolea | | 6.26 ± 0.47 a | 5.03 ± 0.30 a | 5.97 ± 0.52 a | <DL | 4.86 ± 0.14 a |
| Leccino | 7 | 6.10 ± 0.51 a | 2.17 ± 0.24 b | 6.08 ± 0.38 a | <DL | 4.97 ± 1.41 a |
| N. Messinese | | 3.20 ± 0.22 b | 0.00 ± 0.00 c | 4.49 ± 0.41 b | <DL | 0.33 ± 0.65 b |
| Sig. | | ** | ** | ** | n.s. | ** |
| Carolea | | 5.85 ± 0.65 a | 4.96 ± 0.55 a | 5.67 ± 0.71 | <DL | 5.03 ± 0.78 |
| Leccino | 15 | 5.21 ± 0.29 ab | 4.71 ± 0.48 a | 5.51 ± 0.62 | <DL | 3.48 ± 0.91 |
| N. Messinese | | 4.68 ± 0.42 b | 3.60 ± 0.45 b | 5.00 ± 0.58 | <DL | 4.09 ± 0.85 |
| Sig. | | * | ** | n.s. | n.s. | n.s. |
| Carolea | | 5.70 ± 0.18 | 4.61 ± 0.79 a | 5.75 ± 0.29 | <DL b | 4.16 ± 0.80 |
| Leccino | 30 | 5.79 ± 0.10 | 1.92 ± 0.54 b | 5.47 ± 0.21 | 2.91 ± 1.04 a | 4.96 ± 0.48 |
| N. Messinese | | 5.53 ± 0.61 | 4.78 ± 0.94 a | 5.65 ± 0.71 | <DL b | 4.00 ± 1.21 |
| Sig. | | n.s. | ** | n.s. | ** | n.s. |
| Carolea | | 6.25 ± 0.14 a | 6.24 ± 0.42 a | 6.28 ± 0.31 | <DL b | 4.44 ± 1.14 ab |
| Leccino | 60 | 5.99 ± 0.26 a | 2.98 ± 0.22 c | 6.10 ± 0.20 | 3.15 ± 1.07 a | 4.77 ± 0.65 a |
| N. Messinese | | 5.24 ± 0.24 b | 4.76 ± 1.07 b | 5.90 ± 0.93 | <DL b | 2.96 ± 0.88 b |
| Sig. | | ** | ** | n.s. | ** | * |

Data are expressed as means ± standard deviations of log CFU/mL. Different letters indicate statistical differences within the same column and for each sampling time. ** Significance at $p \leq 0.01$; * Significance at $p \leq 0.05$; n.s. not significant. DL = detection limit.

### 3.5. Results of Sensory Analysis

The results of sensory analysis are shown in Figure 2a,b. The obtained QDA profile allows individual quantification of the characteristics of a food product. The final figure is obtained by averaging the intensity values of each descriptor on unstructured scales arranged radially on the graphic plane. By fixing all the meaningful values on the axes and joining them together, we obtained a spider plot representing the sensory profile of the product.

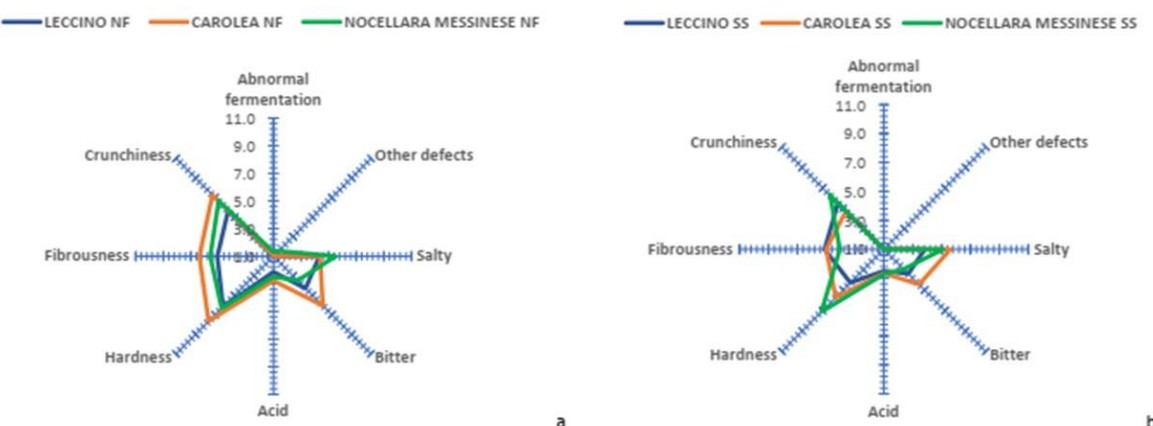

**Figure 2.** Sensory profile of olives processed with natural fermentation (**a**) and in the Spanish style (**b**).

The ANOVA results showed significant differences among the samples of olives in the three cultivars, treated with natural fermentation, for all descriptors. Only N. Messinese showed a slight increase in "other defects" attributable to the taste sensation reminding of soil, but this value does not affect the suitability of this product for commercial markets.

Carolea was evaluated as the most bitter and acidic cultivar and presented a greater hardness, crunchiness and fibrousness than the other cultivars. The Leccino cultivar showed the lowest acidic values; moreover, it showed the lowest kinaesthetic characteristics values. N. Messinese showed a lower level of bitterness and more balanced kinaesthetic descriptors. An ANOVA analysis of the sensory descriptors of the cultivars treated using the Spanish style showed that none of the cultivars developed abnormal fermentations or other defects. Carolea showed the highest salty and bitter values. The N. Messinese cultivar showed the lowest bitter values and presented the best kinaesthetic characteristics. These descriptors are very important commodity parameters for the preparation and marketing of table olives.

## 4. Discussion

The three double-purpose cultivars tested in the present study were particularly different, with N. Messinese drupes being the most suitable as table olives for their flesh-to-pit ratio of 6.4 and according to their high-weight fruits of 9.14 g. The sampled Carolea olives did not reach the best size which can be achieved by this variety [13] but its olives showed a flesh/pit ratio close to the optimal parameter of five and a good weight [23]. Regarding Leccino, this is a cultivar used mainly for oil purposes but, despite the low size and weight of these olives, they are also appreciated as typical table olives [16,24] in different Italian regions. For this reason, Leccino olives are largely considered dual-purpose olives.

The chemical parameters of both treatments (as pH values and sugar consumption in olive pulp) together with the growth of different microbial populations highlighted a different fermentation trend. The long time required by NF results in a lower pH close to the hygienic safety limit (4.17–5.02 range), while the fast fermentation obtained by treating the olives with lye solution results in pH values that are too high (4.86–6.17 range). In both situations, these results highlighted the need to improve fermentation by using a selected starter culture [25] that can drive the fermentation by rapid acidification. Benítez-Cabello et al. [26] used different LAB and a yeast strain, previously isolated from the biofilm of table olives, to inoculate Manzanilla Spanish-style green table olives starting with a pH range of 6.19–6.33 and obtaining a final pH range of 4.08–4.38. The authors concluded that the use of this kind of starter led to a rapid acidification and to enhanced organoleptic characteristics. Regarding the olive phenols, the cultivars already exhibit different phenolic compositions, with the highest values in Carolea olives, as expected, since the Carolea cultivar is known as a source of table olives and extra virgin olive oil particularly rich in phenols [14,15,27]. Benincasa et al. [28] also found Carolea samples with

a higher Verb content than the other analysed cultivars. Moreover, NF and SS methods highlighted major differences in phenolic compounds. In fact, focusing on Carolea, the TyrOH and Verb content was reduced by almost half in olives treated with the SS method. This means that NF olives are a better source of bioactive compounds than SS olives, whose phenolic content is drastically affected by the alkaline treatment and the successive washing steps [29,30]. SS olives are mainly produced to obtain a saleable product in a very short time compared to fermentation as the only debittering method. However, the nutritional quality of these olives is worse than those which are fermented. Moreover, Carolea olives already show a dark grey colour after 2 h immersion in lye solution, probably due to their skin characteristics. This skin behaviour makes these olives unsuitable to be treated with the SS method.

The result of microbiological analyses highlighted a higher presence of yeasts and moulds in the SS samples than in the NF samples, with the highest values in the Carolea and Leccino brine samples. Moreover, the high coliform count found in SS samples once again underlines the hygienic problem with the use of alkali treatment. The mesophilic aerobic counts of SS olives were similar to those reported by Sab et al. [30] in the green Sigoise cultivar during SS industrial processing, while the Enterobacteriaceae in that case dropped down to zero after 20 days. However, coliforms, which belong to the same family, remained at a high count in all the samples analysed in the present study. The cause is likely the high pH value registered up to the end of fermentation. Indeed, it is well known that during the first phase of Spanish fermentation, Gram-negative bacteria prevail; if LAB are not able to induce a decrease in pH, several olive defects can occur [12]. However, according to Chammem et al. [31], the presence of coliform bacteria could be related to the low percentage of NaOH and NaCl applied during SS processing.

Sugar content in olive drupes is lower than any other edible fruit and the major free sugars in fresh olive pulps are glucose, fructose, mannose and galactose. In our study, total sugars, expressed in g/100 g of dry weight (DW), ranged from 11 to 19 for freshly harvested olives (t0) from the Carolea, Leccino and N. Messinese cultivars. Marsilio and co-workers [32] reported a concentration between 2 and 8 g/100 g (DW) for olives from the Ferrandina, Douro, Hojiblanca, Cassanese, Taggiasca and Thasos cultivars. Trapani and co-workers reported changes in the total sugar content of between 8.5 and 18.5 g/100 g (DW) in Moraiolo cultivars [33]. A value of 8 g/100 g (DW) was found for an unusual Tunisian olive variety, Dhokar, which is characterized by the sweet taste of its fruit [34]. Ivancic and co-workers [35] identified five sugars, the most important being glucose, in both the pulp and skin of Leccino olive drupes. A total sugar content between 3.5 and 5 g/100 g (DW) was registered. Such differences in concentrations probably reflect the metabolic behaviour of each cultivar in relation to the genotype and to different climatic and environmental conditions. However, table olives can be considered as practically free of sugar products since the microorganisms present in brines consume sugars during the fermentation process or brine storage. In our experimental trial, at the end of the transformation into table olives, the sugar content in naturally processed Leccino and Carolea samples was practically null. N. Messinese table olives retained 0.6% of their initial values. For SS fermentation, the sugar content was recovered at about 4% of its initial value for Carolea and N. Messinese table olives, and less than 1% for Leccino olives. These results are higher than those found in the literature; in fact, the total soluble sugars, the sum of glucose, fructose and mannitol, found by Lopez-Lopez and co-workers [36] were in the range of 0–0.24 g/100 g (DW). The total soluble sugars found by Iassoui and co-workers [37] for Picholine, Meski and Manzanella naturally fermented olives were 0.39, 0.19 and 0.26 g/100 g (DW), respectively.

The most relevant factors influencing the phenols in olive drupes are the cultivar, together with the growing conditions and the fruit ripening. Fresh pulps are very rich in phenolic compounds, which undergo a series of transformations during the fermentation processes. Both SS and NF debittering methods lead to a significant loss of phenols; a greater loss is caused by the initial lye treatment carried out in the SS method [27,38,39]. The results obtained showed that the variety of olives is a determining factor for the content

of total phenols and the quality of individual phenols in fresh olives which will, therefore, be reflected in table olives. As is widely known from the literature, fresh olive pulps are very rich in phenolic compounds. The values found for Carolea, N. Messinese and Leccino olives were in the range of 7824 to 10,734 mg/kg; these results are in agreement with those found by Baiano and co-workers, who analysed twelve Italian olive cultivars and recorded the highest phenolic content for Peranzana and Cellina di Nardò (14,000 mg/kg) and the lowest for FS17 and Cima di Melfi (7000 mg/kg) [40]. Furthermore, these values are similar to those found for wild olives (*Olea europaea* L.), whose total phenol contents are about 7200 mg/kg [41]. The total phenolic content was higher in Carolea (26,000 mg/kg), Grossa di Gerace (20,000 mg/kg) and N. Messinese fresh olives (23,000 mg/kg) [15], and lower in ten Greek olive cultivars (Koroneiki, Lianolia Kerkyras, Mastoidis, Arbequina, Adramytini, Megaritiki, Gaidourelia, Kalamon, Konservolia and Chalkidiki) whose content ranged between 590 and 1980 mg/kg [42]. However, regardless of the initial heritage of the phenolic compounds, processing table olives decreases their levels.

From our results, olives processed with the SS method suffered a loss of phenols of more than 90% and those subjected to NF, 70%. This trend has also been demonstrated by other authors [43,44]. Table olives' processing decreases the levels of Ole, especially if a lye treatment has been performed, with a concomitant increase in the hydrolysis products TyrOH and Tyr [45,46]. NF olives preserved, in fact, 25% of the Ole initial content, increasing the concentrations of TyrOH and Tyr by 80% and 50%, respectively. These results are higher but follow the same trend as those of other authors [47,48]. A study on Kalamata table olives reported the concentration of single phenols, such as verbascoside, rutin and luteolin, after performing both Spanish and natural fermentation. The concentrations of verbascoside, rutin and luteolin in naturally fermented table olives were 332, 17 and 92 mg/kg, respectively; the lowest values were obtained for Spanish-style table olives [49]. Overall, the obtained data demonstrate that when the olives are ready to be consumed, naturally fermented table olives possess a higher amount of healthy phenolic compounds.

Sensory analysis is very important because it has an important impact on food consumer acceptability. Sensory analysis showed that the N. Messinese cultivar exhibited good kinaesthetic characteristics, and the same result was also highlighted by De Bruno et al., 2019 [15]. Carolea olives, treated with the SS method, show a dark grey colour of the skin, so the olives do not have a good visual perception. Cocolin et al. [49] in their research affirmed that the treatment with NaOH modifies the composition of the table olive ecosystem to a great extent and promotes a fermentation process which has different results, in terms of bacterial species and strain. Furthermore, the debittering process influences the number of species present on the surface of olives. In fact, the cultivars' sensory evaluation showed that the SS olives are less bitter than the same naturally fermented cultivars. It is likely that the treatment with soda has an effect in chemically determining a greater debittering, but at the same time it may modify the product flavour by having modified the bacterial species present.

## 5. Conclusions

This paper presents the results of the transformation into table olives of drupes widely used for virgin olive oil production by using the Spanish-style method (SS) and natural fermentation (NF).

The chemical parameters of both treatments highlighted the need to improve fermentation by using a selected starter culture that can drive the fermentation by rapid acidification. NF olives are a better source of bioactive compounds than SS olives, whose phenolic content is drastically affected by the alkaline treatment and the successive washing steps. If sugar content in olive drupes is lower than any other edible fruit, in table olives it can be considered practically null.

The result of microbiological analyses highlighted a higher presence of yeasts and moulds in the SS samples than in the NF samples, with the highest values in Carolea and

Leccino brine samples. Moreover, the high coliform count found in SS samples once again underlines the hygienic problem with the use of alkali treatment.

The results obtained from the sensory analysis allow us to affirm that N. Messinese and Leccino used for olive oil can also be used to obtain promising table olives. In detail, the sensory analysis showed that N. Messinese olives exhibited better kinaesthetic characteristics with the SS method than the NF method, while the contrary was shown for Leccino olives. On the other hand, Carolea—although suitable as NF table olives—requires too long for natural debittering, and therefore needs a pre-treatment to shorten the fermentation time (pitted or crushed olives). Carolea is not suitable for Spanish-style debittering due to the characteristics of its easily oxidised skin.

**Author Contributions:** Conceptualization, N.T., I.M. and F.V.R.; methodology, N.T., C.A.C.R., C.B., P.F. and F.V.R.; software, N.T., C.B., I.M. and F.V.R.; validation, F.V.R., N.T. and C.B.; formal analysis, N.T., C.A.C.R., R.N. and F.V.R.; investigation, N.T. and S.M.; resources, F.V.R. and I.M.; data curation, N.T., F.V.R., C.B., R.N. and S.M.; writing—original draft preparation, N.T., F.V.R. and P.F.; writing—review and editing, N.T., C.B. and F.V.R. All authors have read and agreed to the published version of the manuscript.

**Funding:** This research was funded by the Italian Ministry of Agriculture, Food and Forestry (MiPAAF), grant designation 'Caratterizzazione e valorizzazione delle olive da mensa e a duplice attitudine'—ALIVE, D.M. 93880 from 29 December 2017.

**Institutional Review Board Statement:** Not applicable.

**Informed Consent Statement:** Not applicable.

**Data Availability Statement:** Not applicable.

**Conflicts of Interest:** The authors declare no conflict of interest.

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
