# Peer review of "Comparing Spanish-Style and Natural Fermentation Methods to Valorise Carolea, Nocellara Messinese and Leccino as Table Olives"

_horticulturae, doi:10.3390/horticulturae9040496_

Round 1
Reviewer 1 Report
Dear Editor, thank you very much for inviting me to review this manuscript.
The manuscript entitled Comparing Spanish style and natural fermentation methods to 2 valorise Carolea, Nocellara Messinese and Leccino as table olives, presents a theme of great relevance, especially for the table olive industry. Currently, most olive tree varieties are used for olive oil extraction. This work aims to reinforce the number of varieties that can be used for table olives.
In my opinion, the article explores an important theme, is well structured and presents interesting results. The authors were careful to use current references to justify their results. However, some shortcomings need to be improved, namely:
Line 20 and 68 – remove the extra. (We only have an extra category after analyzing the oils in the laboratory, so until we know which category you sell, we only call it virgin olive oil).
Material and methods
Washing with tap water and then curing naturally turns out not to be one of the best options, as we are eliminating the microbial load that would be important for fermentation. The ideal would be to use potable water from a well.
Line 83 - what is the maturation index used? They should mention the level of maturation, as this will have an influence on the sensory level.
Was the titratable acidity not determined? The conjugation of pH with acidity would be important to determine actions during the fermentation process, especially natural fermentation.
In table 1 instead of “pit” in my opinion I would call “endocarp”.
Regarding the fat content? It was determined? This parameter can be important when defining the order (for olive oil or table olives).
Texture is a parameter that should have been used to justify the values obtained in the sensory analysis.
The results obtained in the sensory analysis (acidic and salty) seem comparatively low to me. The sensation of saltiness, compared to the brine that was used to ferment the olives, is almost halved. As well as the acid sensation values, since an acidifier was used to lower the pH of the brine.
Author Response
Dear Reviewer, thank you very much for your valuable comments on our manuscript.
Below we list all your remarks together with a point-by-point response to every question.
In the revised version of the manuscript all the revisions are highlighted by using Word Tracking changes.
Reviewer 1
Line 20 and 68 – remove the extra (We only have an extra category after analyzing the oils in the laboratory, so until we know which category you sell, we only call it virgin olive oil).
Reply: Thank you for your annotation. We agree with you. Consequently, the word “extra” was deleted in lines 20 and 68.
Material and methods
Washing with tap water and then curing naturally turns out not to be one of the best options, as we are eliminating the microbial load that would be important for fermentation. The ideal would be to use potable water from a well.
Reply: Thank you for your suggestion. We totally agree with you but in this study we followed the practice usually applied by the companies during table olives production. In some cases, the used water is potable water but not always it is so. The text was not changed.
Line 83 - what is the maturation index used? They should mention the level of maturation, as this will have an influence on the sensory level.
Reply: Thank you for your question. The olives were harvested by the different farmers in agreement with the ALIVE project. So, the harvesting time was decided by the farmers and coincided with the moment when the first turning-colour olives appear. In fact, the olives were all green with less than 5% turning-colour olives. We have added this explanation in line 81.
Was the titratable acidity not determined? The conjugation of pH with acidity would be important to determine actions during the fermentation process, especially natural fermentation.
Reply: Thank you for your suggestion. We agree with you and, indeed, we have made the titratable acidity analysis in some studies. In this case, in addition to pH values, we used the sugar consumption in olive pulp and the microbial growth of different populations as indices of fermentation trend. This was added to lines 408-409.
In table 1 instead of “pit” in my opinion I would call “endocarp”.
Reply: Thank you for your suggestion. However, the olive stone is called pit in several references and above all in “TRADE STANDARD APPLYING TO TABLE OLIVES” Resolution No. RES-2/91-IV/04 of THE INTERNATIONAL OLIVE OIL COUNCIL. So, the word “pit” was not changed in the text. https://www.internationaloliveoil.org/wp-content/uploads/2019/11/COI-OT-NC1-2004-Eng.pdf
Regarding the fat content? It was determined? This parameter can be important when defining the order (for olive oil or table olives).
Reply: Thank you for your comment. The fat content of each of the three olive cultivars were calculated only on fresh harvested olives and it was added in Table 1, together with the carpological characterization of the samples. Lines 96-98 were added to M&M section, and lines 235-237 were added to results. However, the analysed cultivars are usually classified as dual-purpose olives, as written in lines 70-74 of the introduction.
Texture is a parameter that should have been used to justify the values obtained in the sensory analysis.
Reply: Thank you for your suggestion. Texture is an important parameter especially when a panel for sensory analysis is not available. In this case, we preferred to analyse the texture in its qualitative aspects (hardness, fibrousness and crunchiness) by means of internal judges already trained for the sensory assessment of several kind of food.
The results obtained in the sensory analysis (acidic and salty) seem comparatively low to me. The sensation of saltiness, compared to the brine that was used to ferment the olives, is almost halved. As well as the acid sensation values, since an acidifier was used to lower the pH of the brine.
Reply: Thank you for the annotation. However, we can say that factors related to olive characteristics such as amount of pulp and cuticle thickness do not allow a prediction about the amount of saline or lactic acid solution that will be absorbed by the drupe. In addition, the panel members are selected by means of a process implemented in accordance with an international standard according (IOC) and becomes skilled after suitable training and whose performance is objectively evaluated based on rules established beforehand by the panel leader. Specifically, for the training of tasters, standard solutions by aqueous solutions of substances such as sodium chloride and lactic acid are used.

Reviewer 2 Report
GENERAL COMMENTS
Dear Authors, congratulations on your manuscript.
Kind Regards,
Reviewer,
SPECIFIC COMMENTS
The consumption of table olives is very common in some areas of the world. The topic studied by the authors of the manuscript is current and important for the scientific and technological panorama.
In all the manuscript: when writing the brands and countries of production of equipment, utensils, among others, they should indicate the city where they were manufactured.
When authors write values in the text that are in the tables, they should indicate the mean ± SD.
Line 82: I think that after Sicily they should write, respectively.
Line 86: Sigma-Aldrich, country?
Line 92: how was the temperature measured/controlled?
Line 101: is it Torrance?
Line 110: Have the procedures defined in ISO been followed?
Line 183: the tasting panel members/judges/panellists tasted how many samples per session?
Line 226: what is the value considered to be the safety limit (is indicated on lines 241 and 242, but is not here)? What is the bibliographic reference for this?
Table 5: I suggest the authors replace 0.00 ± 0.00 by <DL (Detection limit), but they will have to indicate what the detection limit of the methods is for MSA and CCA.
Line 398: Clarify whether the starter cultures inoculated by the authors were commercial or autochthonous.
Line 427 and 428: The authors should clarify what was said by the author 29. Is the problem related to the low percentages of NaCl and NaOH used in the process? or is it the NaCl and NaOH that will induce coliform bacteria in the olives?
Line 431: “11 to 19” at what point in the production process?
Line 467: Oleo europaea L.
Writing one or two paragraphs of conclusions would be positive for the overall manuscript.
Author Response
Dear Reviewer, thank you very much for your valuable comments on our manuscript.
Below we list all your remarks together with a point-by-point response to every question.
In the revised version of the manuscript all the revisions are highlighted by using Word Tracking changes.
Reviewer 2
In all the manuscript: when writing the brands and countries of production of equipment, utensils, among others, they should indicate the city where they were manufactured.
Reply: Thank you for the suggestion. The requested additions were made in the text (M&M section).
When authors write values in the text that are in the tables, they should indicate the mean ± SD.
Reply: Thank you for the suggestion. However, we have added ± SD in the text only about the carpological analyses, because the mean values frequently appear in result section, so the addition of ± SD would make the text heavy, also because it would be a repetition of the contents of the tables.
Line 82: I think that after Sicily they should write, respectively.
Reply: Thank you. The word was added.
Line 86: Sigma-Aldrich, country?
Reply: The city and country of each brand were added.
Line 92: how was the temperature measured/controlled?
Reply: Thank you for your question. See lines 93-94.
Line 101: is it Torrance?
Reply: Done.
Line 110: Have the procedures defined in ISO been followed?
Reply: Thank you for the question. However, no official microbiological criteria are required for table olive. In addition, the Codex Alimentarius describes the minimum requirements related to hygiene for table olives (Muzzalupo, I. ed., 2012. Olive Germplasm: The Olive Cultivation, Table Olive and Olive Oil Industry in Italy). “At the end of the fermentation, according to the Standard 66–1981 (Rev. 1–1987) of the Codex alimentarius (Anonymous 1987), the product should be free from microorganisms and parasites in amounts representing a hazard to health and should not contain any substance originating from microorganisms in amounts which may represent a hazard to health. Packed fermented olives can contain LAB and yeasts used for fermentation (COI 2004)” (Tofalo, R.; Schiron, M.; Perpetuini, G.; Angelozzi, G.; Suzzi, G.; Corsetti, A. Microbiological and chemical profiles of naturally fermented table olives and brines from different Italian cultivars. Antonie Van Leeuwenhoek 2012, 102, 121–131).
Line 183: the tasting panel members/judges/panellists tasted how many samples per session?
Reply: Thank you for your question. Two samples were tasted at each session. Sensory analysis was carried out in the morning before lunch, period when olfactory-gustatory sharpness is optimal (between 10 a.m. and 12 noon). In addition, the same sample was evaluated in different sessions to monitor the performance of the judges. The number of samples per day was added in line 216.
Line 226: what is the value considered to be the safety limit (is indicated on lines 241 and 242, but is not here)? What is the bibliographic reference for this?
Reply: Thank you for your annotation. We added the appropriate references in the line 243.
-Perricone, M.; Bevilacqua, A.; Corbo, M.R.; Sinigaglia, M. Use of Lactobacillus plantarum and glucose to control the fermentation of “Bella di Cerignola” Table Olives, a traditional variety of Apulian region (Southern Italy). J. Food Sci. 2010, 75, 430–436.
-Foti, P.; Russo, N.; Randazzo, C.L.; Choupina, A.B.; Pino, A.; Caggia, C.; Romeo, F.V. Profiling of phenol content and microbial community dynamics during pâté olive cake fermentation. Food Bioscience 2023, 52, 102358.
Table 5: I suggest the authors replace 0.00 ± 0.00 by <DL (Detection limit), but they will have to indicate what the detection limit of the methods is for MSA and CCA.
Reply: Thank you for the annotation. Zero is a value valid for statistical analysis, but we can replace the values 0.00±0.00 with <DL as suggested and maintain the post-hoc letters. The method for the pathogen detection and enumeration was based on plate count according to manufacturer’s instruction for Mannitol Salt Agar (MSA, Oxoid, Milan, Italy) and Chromogenic Coliform Agar Base (CCA, Bibby Scharlau, Milan, Italy), so the detection limit is the same of every plate count (added in lines 125-127).
Line 398: Clarify whether the starter cultures inoculated by the authors were commercial or autochthonous.
Reply: Thank you for improving the text. The lines 415-416 were added.
Line 427 and 428: The authors should clarify what was said by the author 29. Is the problem related to the low percentages of NaCl and NaOH used in the process? or is it the NaCl and NaOH that will induce coliform bacteria in the olives?
Reply: Thank you for the question. In line 446 it was clarified.
Line 431: “11 to 19” at what point in the production process?
Reply: Thank you for the question. In lines 450-451 it was clarified.
Line 467: Oleo europaea L.
Reply: Done.
Writing one or two paragraphs of conclusions would be positive for the overall manuscript.
Reply: Thank you for the question. The conclusion section was added.

Reviewer 3 Report
This article describes a study to determine the quality indicators of drupes belonging to three Italian cultivars and compares two methods for the production of fermented table olives from them: the Spanish style and the natural fermentation method. As a result of the study, the following parameters and quality indicators of olives obtained by two different methods were determined: carpological analysis, pH of brines, the amount of phenolic compounds, sugar in the pulp, microbiological indicators, organoleic indicators, etc. The advantages and disadvantages of each of the fermentation methods used were discussed.
The article is framed in compliance with the requirements and includes a qualitatively conducted and designed experiment. The proposed methods are valuable for this area of research, but the article has a number of shortcomings.
After reading the article, there were some comments and suggestions to the authors.
1. The authors declare that the aim of the work was to increase the value of the table olives used in the study. However, the authors did not achieve this goal. There is no information on value enhancement in the article, only a good analysis of different fermentation methods, and even based on comparing these different methods, the authors conclude that each of them has its own pros and cons. So what kind of fermentation should ultimately be offered to the olive producer?
2. In the abstract, the authors write that, based on the results obtained, it was emphasized that it is necessary to improve fermentation through the use of a selected starter that can stimulate fermentation through rapid acidification. However, there is no information in the article about which starter the authors chose and how its introduction affected the quality of the olives. There is a small piece of text in the discussion, where the authors again talk about choosing a starter and refer to source 23, but there is no more information about this in the article. Did the authors use leaven? Where are the quality indicators of olives prepared using sourdough?
3. There is very little information in the discussion about which fermentation technology should be chosen in the end? And there is no clear conclusion. The authors simply compare different methods and point out their advantages and disadvantages. It seems that the authors conducted many experiments, but did not come to a consensus. In my opinion, it is necessary to give clear recommendations on the parameters and method of fermentation based on the large amount of data that was obtained by the authors as a result of the study.
Author Response
Dear Reviewer, thank you very much for your valuable comments on our manuscript.
Below we list all your remarks together with a point-by-point response to every question.
In the revised version of the manuscript all the revisions are highlighted by using Word Tracking changes.
Reviewer 3
- The authors declare that the aim of the work was to increase the value of the table olives used in the study. However, the authors did not achieve this goal. There is no information on value enhancement in the article, only a good analysis of different fermentation methods, and even based on comparing these different methods, the authors conclude that each of them has its own pros and cons. So what kind of fermentation should ultimately be offered to the olive producer?
Reply: Thank you for the suggestion. We have better clarified the requested part in the conclusions, that was added to manuscript.
- In the abstract, the authors write that, based on the results obtained, it was emphasized that it is necessary to improve fermentation through the use of a selected starter that can stimulate fermentation through rapid acidification. However, there is no information in the article about which starter the authors chose and how its introduction affected the quality of the olives. There is a small piece of text in the discussion, where the authors again talk about choosing a starter and refer to source 23, but there is no more information about this in the article. Did the authors use leaven? Where are the quality indicators of olives prepared using sourdough?
Reply: Thank you for the questions. In this manuscript (MS) we did not use any starter culture, even if we have several publications focusing on this issue. The activity of the MS arises from a project whose focus was to help the table olive farmers in the choice of the best method for each cv. The need to standardise and industrialise the processing best suited to each cultivar could economically improve the regional productions. We have mention the microbial starters only with the purpose of underlining the need to improve the fermentation quality (in NF method) by using a selected starter culture. We are strongly convinced that the use of starters will be the future of this kind of fermented products.
- There is very little information in the discussion about which fermentation technology should be chosen in the end? And there is no clear conclusion. The authors simply compare different methods and point out their advantages and disadvantages. It seems that the authors conducted many experiments, but did not come to a consensus. In my opinion, it is necessary to give clear recommendations on the parameters and method of fermentation based on the large amount of data that was obtained by the authors as a result of the study.
Reply: Thank you for the suggestion. As for question n. 1, we have better clarified the requested part in the conclusions, that was added to manuscript.

Round 2
Reviewer 2 Report
Dear authors,
Thank you for your answers to my questions.
Kind regards,
Reviewer
Reviewer 3 Report
Thanks to the authors for providing answers and corrected conclusions.